# Sperm Toolbox—A selection of small molecules to study human spermatozoa

**Franz S. Gruber**[1]☯, **Anthony Richardson**[2]☯, **Zoe C. Johnston**[3]☯, **Rachel Myles**[3], **Neil R. Norcross**[2], **David P. Day**[2], **Irene Georgiou**[2], **Laura Sesma-Sanz**[1], **Caroline Wilson**[2], **Kevin D. Read**[2], **Sarah Martins da Silva**[2], **Christopher L. R. Barratt**[3], **Ian H. Gilbert**[2], **Jason R. Swedlow**[1]*

1 Divisions of Computational Biology and Molecular, Cell and Developmental Biology, and National Phenotypic Screening Centre, School of Life Sciences, University of Dundee, Dundee, United Kingdom, 2 Drug Discovery Unit, Division of Biological Chemistry and Drug Discovery, School of Life Sciences, University of Dundee, Dundee, United Kingdom, 3 Division of Systems Medicine, School of Medicine, Ninewells Hospital and Medical School, University of Dundee, Dundee, United Kingdom

☯ These authors contributed equally to this work.
* jrswedlow@dundee.ac.uk

## Abstract

Male contraceptive options and infertility treatments are limited, and almost all innovation has been limited to updates to medically assisted reproduction protocols and methods. To accelerate the development of drugs that can either improve or inhibit fertility, we established a small molecule library as a toolbox for assay development and screening campaigns using human spermatozoa. We have profiled all compounds in the Sperm Toolbox in several automated high-throughput assays that measure stimulation or inhibition of sperm motility or the acrosome reaction. We have assayed motility under non-capacitating and capacitating conditions to distinguish between pathways operating under these different physiological states. We also assayed cell viability to ensure any effects on sperm function are specific. A key advantage of our studies is that all compounds are assayed together in the same experimental conditions, which allows quantitative comparisons of their effects in complementary functional assays. We have combined the resulting datasets to generate fingerprints of the Sperm Toolbox compounds on sperm function. The data are included in an on-line R-based app for convenient querying.

## Introduction

One of the grand challenges for science and society is the development of novel contraceptives that are safe and deliver control of reproduction to men and women all over the world. In parallel, a deeper understanding of sperm maturation, motility, fertilization, and possible interventions for male infertility is required to counter declining birth rates. Recently, the application of advanced technologies has helped move the human fertility field forward, revealing new potential targets for both contraception and fertility, and new compounds as contraceptive candidates (reviewed by [1–3]). A key part of this effort is the development of mechanistic insight and drug candidates that affect the behaviour and properties of human

**Data Availability Statement:** All process data are included in the Supplemental Info. Data have also been deposited at zenodo, and are available at

https://doi.org/10.5281/zenodo.10156888 (DOI:
10.5281/zenodo.10156888).

**Funding:** The authors acknowledge support by the
Bill & Melinda Gates Foundation (INV-007117). The
funders had no role in study design, data collection
and analysis, decision to publish, or preparation of
the manuscript.

**Competing interests:** The authors have declared
that no competing interests exist.

spermatozoa [4–7]. Towards this end, we have recently developed an automated live human
sperm, target-agnostic, phenotypic screening platform that can profile the effects of 1000s of
compounds on spermatozoa. These parameters are then used to identify modifiers of sperm
function [8, 9]. These modifiers can cause either positive or negative effects and therefore are
starting points for either fertility treatments (including medical assisted reproductive technologies) or contraceptive development.

A limitation of the phenotypic assays we have employed is the lack of mechanistic information on compounds that score in the assay. Because there is so little known about the molecular
mechanisms that control sperm motility, we are unable to develop hypotheses of potential targets when new hits are identified in the sperm motility assay. However, it is now well-established that features derived from images in phenotypic assays, when compared to phenotypes
measured on reference compounds, can accurately stratify genetic or chemical perturbations
in phenotypic screens [10, 11]. We have therefore collected 83 reference compounds that have
been reported in the literature to affect sperm function in various assays as well as compounds
we have observed to modify sperm function in our own assays, to form the Sperm
Toolbox (STB) (Table 1). In constructing the STB, we have focussed on compounds reported
in the literature that affect human systems, but compounds tested in relevant animal models,
cell-based, or biochemical systems have also been included (Fig 1A). Published analyses of the
compounds in the STB have indicated effects on sperm function e.g. motility, capacitation,
acrosome reaction, $Ca^{2+}$ signalling, viability, or related to other processes (e.g. spermatogenesis, liquefaction, or ejaculation; Fig 1C). Compound annotations are included in the dataset
derived from the STB and indicate that Enzymes/Kinases, Receptors, Channels and Transporters are the most represented target of the STB compound classes (Fig 1D). Similar toolboxes
have been established for other diseases or targets, e.g. the MMV Pandemic Response Box [12],
the SGC Epigenetic Chemical Probe Collection [13], or the Drug Repurposing Hub [14].

List of compounds included in Sperm Toolbox. Published Phenotypes are summaries of
reported effects on sperm function related to other processes (e.g. spermatogenesis, liquefaction,
or ejaculation). Compounds from our own screening campaigns, which have not been published and are included into the Sperm Toolbox have been labelled with 'This study' (Fig 1B).

To demonstrate the value of the STB, we have assayed STB compounds in a series of high
throughput phenotypic assays of sperm function (Fig 1A). The resulting quantitative fingerprint of sperm function modifiers provides a useful reference for drug discovery campaigns as
well as experiments that explore relevant molecular mechanisms. An advantage of using a
high-throughput platform is the ability to simultaneously compare many compounds under
the same assay conditions (same buffer system, same sample), within a short amount of time.
This overcomes limitations of screening compounds at low throughput, where only few compounds can be tested with the same donor pool, one compound at a time. Invariably some of
the compounds we have included in the STB have proven to have little effect in human sperm,
at least under the conditions and at the concentrations we have used. We have maintained
these compounds in the STB and reported the results to ensure the published literature is correct and authoritative.

## Results and discussion

We tested the STB compounds in multiple functional sperm assays, as well as more general
cytotoxicity and aqueous solubility assays (Fig 1A). Using information from the Drug Repurposing Hub [14], we generated a protein-protein interaction network of targets related to STB
compounds (Fig 1E). This network consists of multiple subclusters related to functions in spermatozoa (Fig 1F) and other biological processes, functions, and pathways (Fig 1G).

**Table 1. Sperm Toolbox compounds.**

| STB Number | Name | Published Phenotype | References |
|---|---|---|---|
| 1 | ASI-8 | Blocks soluble adenyl cyclase (sAC) | [15–17] |
| 2 | bpV(phen) | Blocks progesterone induced acrosome reaction; blocks progesterone induced $Ca^{2+}$ response | [18] |
| 3 | Tideglusib | Spermicidal; vaginal treatment | [19] |
| 4 | Genistein | Blocks progesterone induced acrosome reaction; blocks progesterone induced $Ca^{2+}$ response | [18] |
| 5 | T16Ainh-A01 | Blocks acrosome reaction induced by recombinant ZP3 | [20] |
| 6 | Niflumic acid | Blocks acrosome reaction induced by recombinant rhZP3 | [20] |
| 7 | Hexachlorophene | Reduces sperm motility; blocks sAC | [15] |
| 8 | Fenvalerate | Blocks progesterone induced acrosome reaction | [18] |
| 9 | Niclosamide | Shape abnormality | [21] |
| 10 | BMS-189453 | Interferes with spermatogenesis | [22] |
| 11 | Dipyridamole | Enhances sperm motility | [23] |
| 12 | Cloperastine hydrochloride | Reduces sperm motility | This study |
| 13 | Resiquimod | Reduction of sperm motility; only sperm with X chromosome | [8, 24] |
| 14 | Compound B4 | Reduces sperm motility | [25] |
| 15 | 2-Guanidinobenzimidazole | Reduces sperm motility | [26, 27] |
| 16 | Alexidine dihydrochloride | Reduces sperm motility | [8] |
| 17 | TP 003 | Enhances sperm motility | [9] |
| 18 | NSC 697923 | Reduces sperm motility | This study |
| 19 | Pha-665752 | Modulates P4 evoked $Ca^{2+}$ response | [28] |
| 20 | Ouabain | Reduces sperm motility | [29] |
| 21 | Trequinsin hydrochloride | Increases hyperactivated sperm; increases penetration of sperm into viscous media, increases motility | [28] |
| 22 | Leelamine hydrochloride | Modulates P4 evoked $Ca^{2+}$ response | [28] |
| 23 | SLM 6031434 hydrochloride | Induces acrosome reaction | This study |
| 24 | JQ1 | Interferes with spermatogenesis; reduces sperm count; reduces motility | [30] |
| 25 | n-[4-(4-Chlorophenyl)-1,3-thiazol-2-yl] guanidine | Related to Hv1 | [27] |
| 26 | Nafamostat | Interferes with liquefaction; reduces sperm motility | [31, 32] |
| 27 | Anandamide | Reduces sperm motility | [26] |
| 28 | Torin 2 | Enhances sperm motility | [9] |
| 29 | PD 146176 | Antagonizing negative effects of ROS on sperm function such as acrosome reaction | [33] |
| 30 | Mibefradil | Blocks progesterone induced $Ca^{2+}$ response | [34] |
| 31 | CYM 5442 hydrchloride | Reduces sperm motility; induces acrosome reaction | This study |
| 32 | MJ33 | Prevents capacitation-associated tyrosine phosphorylation by PKA | [35] |
| 33 | HC-056456 | Blocks $Ca^{2+}$ and $Na^+$ entry; blocks onset of hyperactive motility | [36] |
| 34 | Urinastatin | Interferes with liquefaction; reduces sperm motility | [31] |
| 35 | Pristimerin | Modulates sperm motility; reduces progesterone induced $Ca^{2+}$ response | [37, 38] |
| 36 | CZC 25146 | LRRK2 neuronal toxicity | [39] |
| 37 | NNC 55–0396 dihydrochloride | Blocks ccl-20 induced $Ca^{2+}$ response; Modulates CatSper | [34] |
| 38 | 666–15 | Reduces sperm motility; induces acrosome reaction | This study |
| 39 | Phenanthroline | Inhibit proteolytic activity of PSA | [32, 40] |
| 40 | CCR6 inhibitor | Patent | [41] |
| 41 | Oprea1_544341 | Reduces sperm motility; role of glycolysis in sperm motility | [42, 43] |
| 42 | A23187 | Induces acrosome reaction | [44] |
| 43 | LRRK2-IN-1 | Enhances | [9] |
| 44 | Disulfiram | immotile sperm; spermicidal | [8, 45, 46] |
| 45 | Forskolin | Stimulates acrosome reaction | [47] |

*(Continued)*

**Table 1.** (Continued)

| STB Number | Name | Published Phenotype | References |
|---|---|---|---|
| 46 | Alfuzosin | Interferes with ejaculation | [48] |
| 47 | Compound 19 | Blocks TSSK; similarity to brigatinib | [49] |
| 48 | IBMX | Modulates capacitation | [35] |
| 49 | Tamsulosin | Interferes with ejaculation | [48] |
| 50 | Imiquimod | Reduction of sperm motility; only sperm with X chromosome | [24] |
| 51 | Bithionol | Blocks sAC | [15, 17] |
| 52 | GF 109203X | Blocks progesterone induced acrosome reaction; blocks progesterone induced $Ca^{2+}$ response | [18] |
| 53 | CTFRinh-172 | Blocks progesterone induced acrosome reaction; blocks sperm hyperactivation; blocks recombinant ZP3 induced acrosome reaction | [50] |
| 54 | Sodium orthovanadate | Blocks progesterone induced acrosome reaction; blocks progesterone induced $Ca^{2+}$ response | [18] |
| 55 | FPL 64176 | Modulates evoked $Ca^{2+}$ response | [28] |
| 56 | 183 | Reduces acrosome reaction | [51] |
| 57 | Esomeprazole | Reduces sperm motility | [52] |
| 58 | 68853496 | Patent; sAC inhibitor | [53] |
| 59 | Gossypol | Causes azoospermia | [54] |
| 60 | Clofarabine | Enhances sperm motility | [9] |
| 61 | B07 hydrochloride | Reduction of sperm motility | [55] |
| 62 | KT 5720 | Evokes $Ca^{2+}$ signaling in sperm | [56] |
| 63 | Linsitinib | enhances sperm motility | [9] |
| 64 | Tafenoquine | Reduces sperm motility; induces acrosome reaction | This study |
| 65 | Hesperadin hydrochloride | Reduces sperm motility | This study |
| 66 | UK 78282 hydrochloride | Modulates $Ca^{2+}$ response | [28] |
| 67 | Tacrolimus | Blocks progesterone induced acrosome reaction | [18] |
| 68 | BPO-27 | CTFR inhibitor patent | [57] |
| 69 | Tak-063 | Enhances sperm motility | [9] |
| 70 | LRE1 | Prevents hyperactivation | [58] |
| 71 | TAE226 | Structural similarity with brigatinib (reduces motility) | This study |
| 72 | GW 843682X | Enhances sperm motility | [9] |
| 73 | GSK1904529A | Inhibition of hyperactivation | [59] |
| 74 | AZD 7762 hydrochloride | Reduces sperm motility | This study |
| 75 | Cyclosporin A | Blocks progesterone induced acrosome reaction | [18] |
| 76 | NVP-AEW541 | Inhibition of hyperactivity | [59] |
| 77 | PF 431396 | Blocks capacitation | [60] |
| 78 | JX 401 | Modulates $Ca^{2+}$ response | [28] |
| 79 | H 89 dihydrochloride | Evokes $Ca^{2+}$ signaling in sperm cells | [56] |
| 80 | Ro 106–9920 | Reduces sperm motility | This study |
| 81 | CCR6 inhibitor 1 | Potentially blocks acrosome reaction | [34, 61] |
| 82 | Herbimycin a | Blocks progesterone induced acrosome reaction; blocks progesterone induced $Ca^{2+}$ response | [18] |
| 83 | Miglustat | Interferes with spermatogenesis | [62] |

To compare the effects of STB compounds on specific sperm functions (e.g. motility, acrosome reaction, viability), we screened them in multiple phenotypic functional assays described below. Each sperm function assay has been repeated in multiple donor pools (see Materials and methods), where each pool consists of cells donated from at least 3 different donors to address donor to donor variability. Each compound has been tested at two concentrations, 10 and 30 µM, and normalized to vehicle controls (DMSO, see Materials and methods). We have

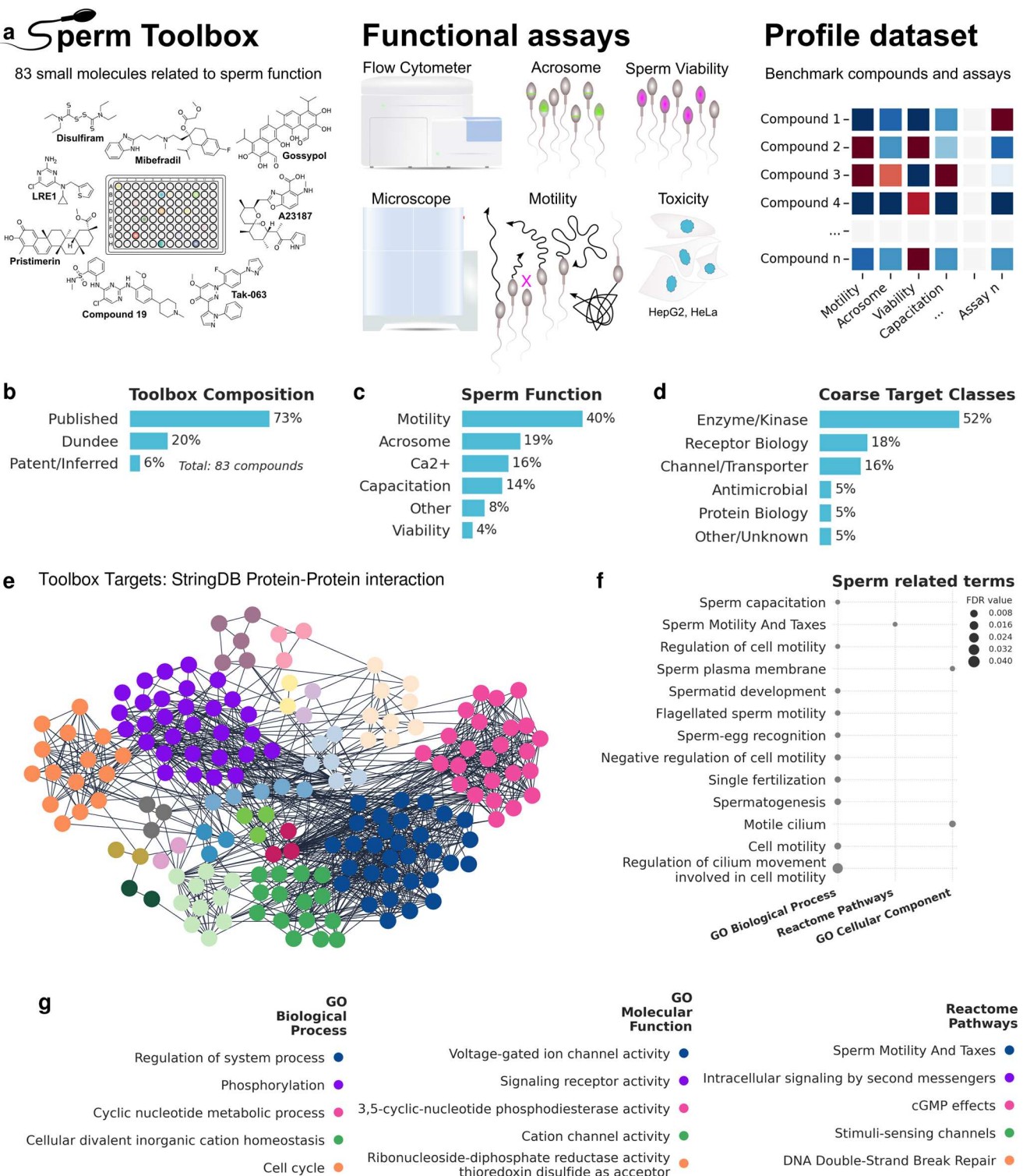

**Fig 1. Design and assay of the Sperm Toolbox.** (A) The Sperm Toolbox is a collection of 83 small molecules with published, observed or inferred effect on multiple biological processes related to the biogenesis or function of spermatozoa. Human sperm cells have been tested in various assays, which allows comparison of compounds under the same experimental conditions. (B) Summary of the STB compound compositions. (C) Summary of annotated (i.e. published, observed, inferred) sperm function of Toolbox compounds. (D) Coarse classification of STB target classes. (E) StringDB Protein-Protein interaction networks (Full network, confidence 0.7) of putative targets of STB compounds. Protein targets have been inferred from the repurposing hub. Network has been clustered using MCL clustering in Cytoscape. (F) Sperm related terms (GeneOntology, Reactome) within the network shown in (E).

Marker size indicates false discovery rate (FDR). (G) Enriched terms within the biggest subclusters in (E). Redundant terms have been filtered using Cytoscape (cutoff 0.5). Color matches subcluster color in (E).

modified our assays from the previously published workflow [8] by increasing incubation time and introducing conditions that support capacitation, which is a sperm maturation process required for fertilization. These changes expand the range of targets that we can explore in our assays. Our live-sperm-based high-throughput motility assays use an automated microscope to record short timelapse movies. These assays can be run in different protocols (S1 Fig). The first protocol, denoted 'motility assay', uses conditions that do not support capacitation and model the state of spermatozoa after ejaculation. In this protocol, cells are incubated with compounds for 10 min prior to the first motility read-out. A second and third motility read-out were performed after 3h and 6h incubation. The second protocol, denoted 'capacitation assay', utilises conditions that support capacitation (S1 Fig). In this protocol, cells are kept in conditions supporting capacitation for 30 minutes prior to 30 minutes incubation with compound, followed by motility read-out. Both protocols allow sperm motility to be measured (0.5 sec time-lapse movies allow tracking of cells and calculation of sperm kinematics). In addition, the capacitation assay also scores hyperactivation (i.e. distinct movement patterns concomitant with capacitation which are crucial for fertilization) (Fig 2A).

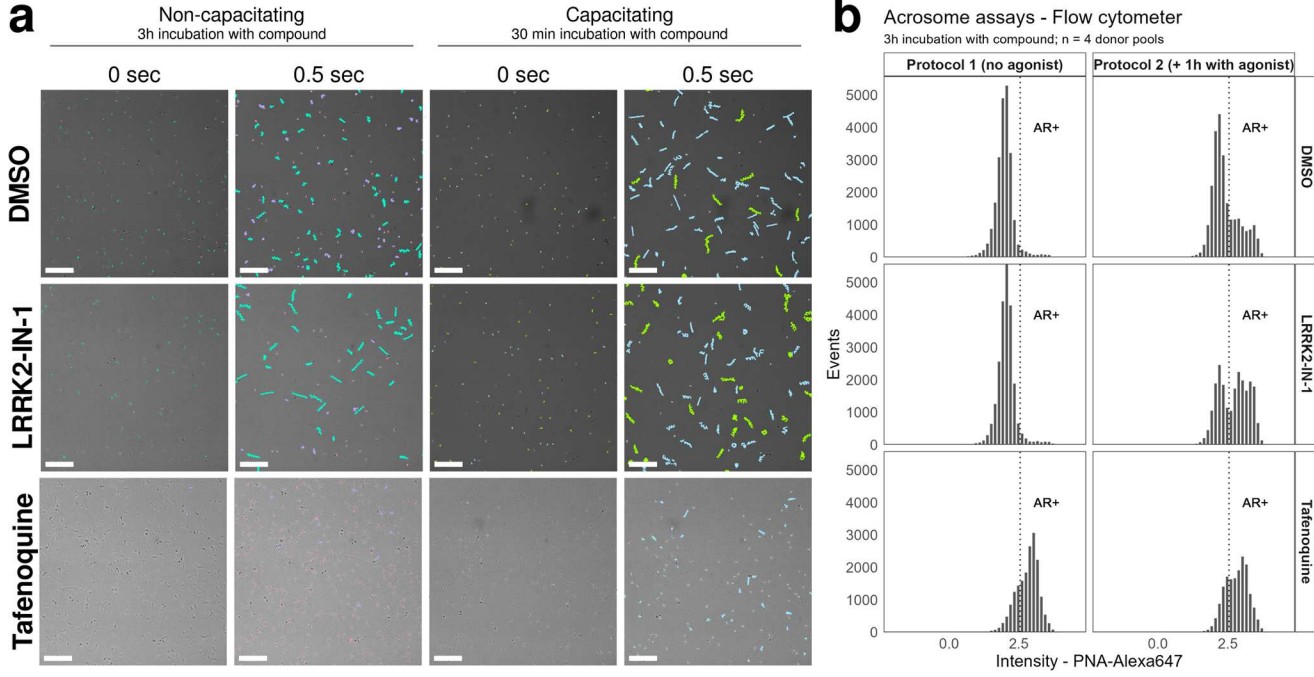

**Fig 2. Example assay data from STB compounds.** (A) Motility assay example images. Tracked sperm cells treated with DMSO (top row), LRRK2-IN-1 (middle row) or Tafenonquine (bottom row). First (0 sec) and last frame (0.5 sec) of time-lapse movies are shown. Non-capacitating conditions (3h incubation with compound) are shown next to capacitating conditions (30 min incubation with compound). Compound concentration was 10 uM for all shown conditions. Colors in the non-capacitating assay indicate immotile sperm (pink), non-progressively motile sperm (lavender) or progressively motile sperm (teal). Colors in the capacitation assay indicate hyperactive sperm (green) or non-hyperactive sperm (blue). Scale bars 50 um. (B) Acrosome assay examples. Histogram depicting number of events and shift in intensity in the PNA acrosome channel (AR+ population). Acrosome assay protocol 1 (3h incubation with compound, no agonist addition; left panel) and acrosome assay protocol 2 (3h incubation with compound, 1h with agonist; right panel). Comparing DMSO (top row), LRRK2-IN-1 (middle row) and Tafenoquine (bottom row). Compound concentration was 10 mM for all shown conditions. Dashed line indicates gate (AR+ population).

The acrosome assays utilize a high-throughput flow cytometer (S1 Fig) to score for compounds inducing the acrosome reaction, using peanut agglutinin conjugated to a fluorescent dye to label the outer acrosomal membrane. We have developed two different screening protocols, named 'acrosome 1' and 'acrosome 2' (S1 Fig). Both protocols are capable of measuring induction of the acrosome reaction upon compound incubation. However, in the acrosome 2 assay, a mixture of physiological agonists of the acrosome reaction is used after cells have been incubated with compounds for 3h under conditions that support capacitation [63]. This addition allowed measurement of inhibition of acrosome reaction (Fig 2B).

We also tested the effect of STB compounds on sperm viability using a complementary live-cell flow-cytometer approach and propidium iodide, which does not permeate intact cell membranes (S1 Fig). In this assay, cells are incubated for 3h with compound in non-capacitating media, prior to viability assessment.

Finally, we also profiled each STB compound for general cytotoxicity in two cell-lines (HeLa and HepG2). The HeLa-based assay uses the cell painting assay [11], while the HepG2 assay utilizes a resazurin based viability read-out [64]. To ensure usability of STB compounds, we also measured aqueous solubility. An overview of additional assays is given in S1 Fig.

This multi-assay compound profiling dataset reveals a broader picture of the action of individual compounds than has been available before and helps find phenotypic similarities induced by compounds. Using unsupervised clustering we observed clusters of compounds with a significant reduction of sperm motility and/or increased amounts of acrosome reacted cells (e.g. Alexidine dihydrochloride, Tafenoquin or A23187) (Fig 3A). Some of these compounds also show significant levels of cytotoxicity in HeLa and/or HepG2 cells as well as decreased sperm viability suggesting that their effect on motility and acrosome reaction is due to toxic effects on spermatozoa. However, other clusters show a significantly increased acrosome reaction and hyperactive motility but do not show effects on sperm viability or general cytotoxicity (e.g. LRRK2-IN-1, Clofarabine, Linsitinib, Trequinsin). One example, LRRK2-IN-1, which enhances motility (progressive and hyperactivated), and increases the number of acrosome reacted cells in our acrosome assay 2 (Fig 2) has been described as a selective inhibitor of LRRK2 kinase, which is involved in Parkinson's disease [65]. This compound shares some structural similarities with two kinase inhibitors in the STB, which decrease sperm motility: Compound 19, described as a potent Testis-Specific Serine/Threonine Kinase 2 (TSSK2) inhibitor [49] and CZC 25146, annotated as a potent LRRK2 inhibitor (Fig 3B and 3C). These three compounds show different activity profiles in our profiling assays, suggesting they may target different pathways in sperm. Interestingly, LRRK2-IN-1 only increases acrosome reacted cells in conjunction with the agonist cocktail (acrosome assay 2), while without there is no change, highlighting the potential of our assays to pick up different effects of the same compounds, when the assay condition has been slightly modified (Fig 2B).

Interestingly, we observed that a number of compounds show weak or no effects in our assays (S2 Fig; grey datapoints). Given these compounds were sourced from publications, patents, and screening systems that mostly report activity in sperm, we have retained them in the toolbox as they may score in other assays or at other concentrations. Alternatively if they do not score in assays of sperm function, their inclusion serves to correct previous results. Either way, this highlights a key strength of our approach. The STB, and the use of high-throughput assays allows the profiling of compounds using the same assay conditions (i.e. buffer system, incubation time and screening concentration) on a large number of human spermatozoa, thus providing reproducible, quantitative assay results. This combination also facilitates the comparison of the effects of the STB compounds between different physiological states (S1 Fig), in a standardised, rapid, and quantitative manner that has not been previously possible in the field of sperm biology. This standardized approach provides

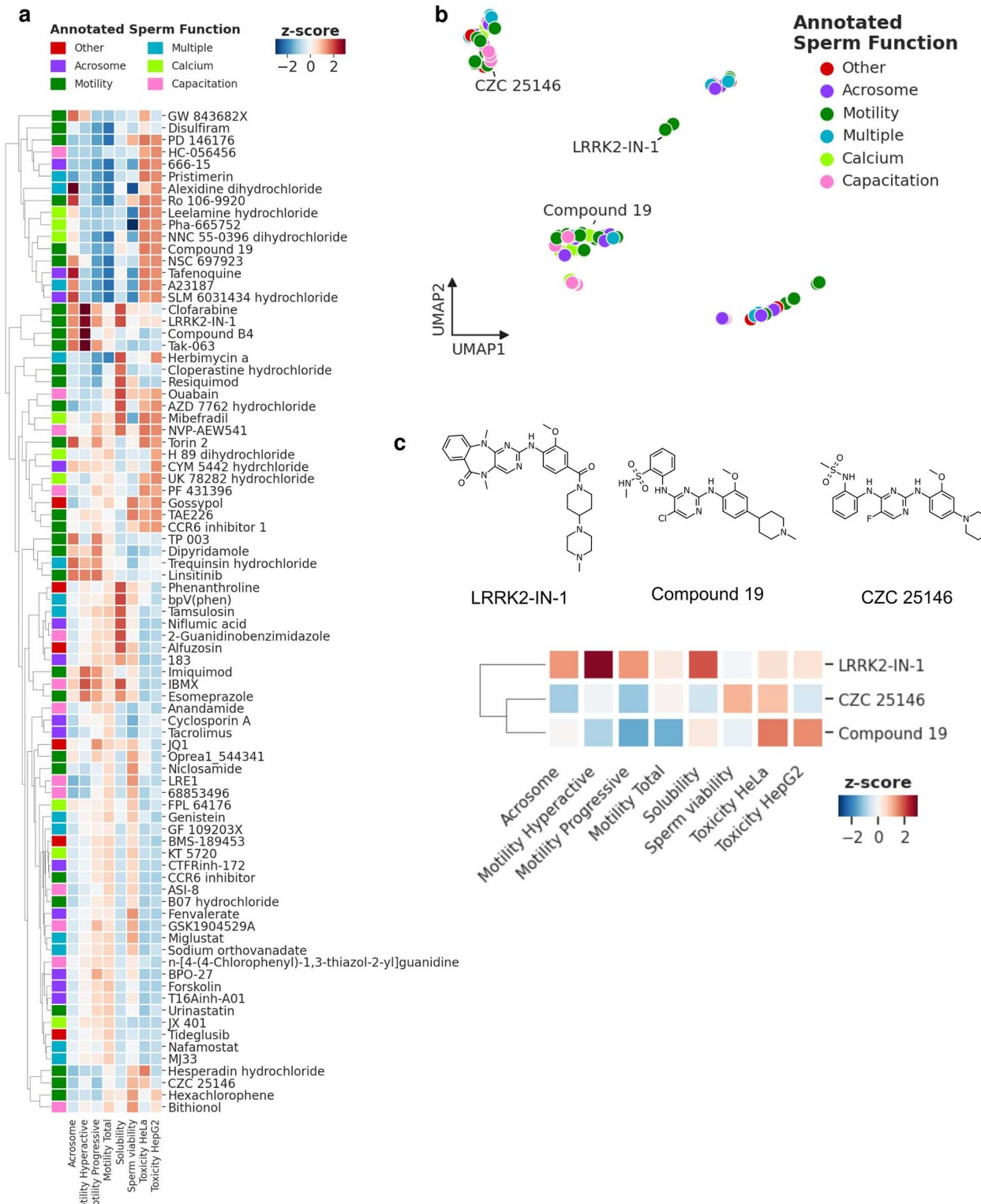

**Fig 3. Visualization of assay results on STB compounds.** (A) Heatmap showing standardized results (z-score) for every Toolbox compound screened at 30 μM in selected assays: Acrosome 2 (3h with compound then induction with acrosome reaction using an agonist mix), Motility Hyperactive (30 min capacitation then 30 min incubation with compound), Motility Progressive (3h incubation with compounds under non-capacitating conditions), Motility Total (3h incubation with compound under non-capacitating conditions)), Solubility (aqueous solubility of compounds after 24h incubation), Sperm viability (3h with compound then propidium iodide assay), Toxicity HeLa (24h compound incubation

then cell painting assay), Toxicity HepG2 (70h incubation with compound then resaruzin based assay). Heatmap color indicates decrease (blue) or increase (red). (B) UMAP plot showing groups of compounds with similar assay profile. Color indicates annotated sperm function. (C) Structures and zoom-in of three compounds with related structures described to modulate LRRK2 kinase but appear to have different effects on sperm function.

an excellent starting point to establish control compounds and assay conditions for novel drug discovery campaigns related to contraception and infertility. We provide the complete dataset (see Supporting information) as well as web interface for data analysis (https://spermtoolbox.shinyapps.io/spermtoolbox/).

## Conclusion

This work represents the first version of the STB. We aim to release annual updates with newly published compounds that modulate the function of spermatozoa. In our next release we aim to include compounds that were published during the collection of the data for this first release e.g., a sAC inhibitor TDI-11861 [4], a CatSper inhibitor RU1968 [66], a SLO3 inhibitor [67] or other contributions from the research community. The STB is a reference for compounds that modulate sperm function and can become an important resource for benchmarking new assays and new compounds. The STB allows comparison of compounds side-by-side using the same assay conditions therefore mitigating the effects of different conditions e.g., buffers, cell populations which may give erroneous results. We aim to use the STB as a reference in our screens and to define as many factors targeted by STB compounds as possible that are critical for sperm function.

## Materials and methods

### Ethical approval

Written consent was obtained from each donor in accordance with under local ethical approval (University of Dundee, SMED REC 20/45). Donors for the study were sourced from July 1, 2020 until 31 March, 2023.

### Sperm handling

Donated human spermatozoa with normal semen characteristics (concentration, motility; WHO 2010) from healthy volunteers with no known fertility problems were used in this study. Samples were obtained by masturbation after $\geq$ 48 h of sexual abstinence, liquefied for 30 min at 37˚C, and processed using density gradient centrifugation, to separate cells using 40/80% Percoll (Sigma Aldrich, UK) fractions. Every experiment has been performed on pooled spermatozoa from $\geq$ 3 donors each.

For experiments in non-capacitating conditions, we used a buffer system slightly modified from [23] using Minimal Essential Medium Eagle (Sigma), supplemented with HEPES (1 M solution, Gibco), sodium lactate (DL-solution, Sigma), sodium pyruvate (100 mM solution, Gibco) and bovine serum albumin (7.5% solution, Sigma).

For experiments in capacitating conditions, we used a buffer system slightly modified from the HTF from [63] to a final composition of 97.8 mM NaCl, 4.69 mM KCl, 0.2 mM MgSO4, 0.37 mM KH2PO4, 2.04 mM CaCl2, 0.33 mM Na-pyruvate, 12.4 mM Na-lactate, 2.78 mM glucose, 21 mM HEPES, 25 mM NaHCO$_3$ and 3 mg/mL BSA. All buffer components were supplied by Sigma-Aldrich unless otherwise stated and all buffer systems were adjusted to pH 7.4.

## Compound handling

Compounds were purchased from vendors indicated in the source publications. 10 mM stock solutions in DMSO were generated. Quality control of each compound has been performed (HPLC-MS and NMR).

Compound plates were generated using 10 mM stock solutions of each STB compound in Echo 384-LDV plates (LP-0200). Prior to running experiments, assay ready plates have been generated using Echo acoustic dispensers (550/555). Each compound was screened at 10 and 30 μM final concentration. Plate types used for assays: Motility, HeLa cell painting and Sperm viability (PerkinElmer 384-well PhenoPlate), Acrosome (Greiner 384-PP V-bottom plates).

## Motility assays

**Motility assay (Non-capacitating).** Cells were rested for 1 h post density gradient centrifugation. 10 μL assay buffer was dispensed into assay ready plates, plates were put onto a shaker instrument for 30 seconds using 1,000 rpm, followed by a short pulse centrifugation. Plates were incubated for 15–30 mins at 37°C prior to addition of 10 μL of pooled cells (at 1 M/mL concentration). Plates were then incubated in a Yokogawa CV7000 microscope set at 37°C for 10 min prior to imaging. Imaging a 384-well plate requires $\leq$ 20 min using the following settings: 2 positions per well, 0.5 sec timelapse movies (3x binning; 11 ms exposure time, 22 ms interval time, 500 intervals). A second and a third timepoint was recorded after 3 h, and 6 h incubation at 37°C. Data has been processed as described in [8]. Assay results are from 5 separate donor pools.

**Capacitation assay.** Cells were capacitated for 30 min at 37°C and 5% $CO_2$. 10 μL assay buffer was dispensed into assay ready plates, plates were put onto a shaker instrument for 30 seconds using 1,000 rpm, followed by a short pulse centrifugation. Plates were incubated for 15–30 mins at 37°C and 5% $CO_2$ prior to addition of 10 μL of pooled cells (at 1 M/mL concentration). This was followed by a 20 min compound incubation step at 37°C and 5% $CO_2$. Plates were then incubated in our CV7000 microscope set at 37°C and 5% $CO_2$ for 10 min prior to imaging. Imaging a 384-well plate requires $\leq$ 20 min using the following settings: 2 positions per well, 0.5 sec timelapse movies (3x binning; 11 ms exposure time, 22 ms interval time, 500 intervals). Data has been processed as described in [8]. Assay results are from 8 separate donor pools.

**Acrosome assays.** In both experimental protocols, 20 μL of pooled cells (at 1 M/mL concentration) were capacitated for 3 h in presence of compounds at 37°C and 5% $CO_2$. In protocol a, cells were then stained with PNA-Alexa647 (ThermoFisher, 1 mg/mL, final dilution 1:1000) and Hoechst dye for 30 min at 37°C and 5% $CO_2$, followed by 10 min fixation at RT by adding 20 μL of 4% paraformaldehyde (Sigma) in PBS. In mode b, a cocktail of progesterone, prostaglandin A and $NH_4Cl$ was added to all compound wells and incubated for an additional 1 h at 37°C and 5% $CO_2$ to induce acrosome reaction. This was followed by fixation and staining protocol as for mode a. In both protocols fixatives and stains were removed and cells were resuspended in PBS, using a 405 platewasher. Plates were then measured on a Novocyte Advanteon high-throughput flow cytometer. Gating of data has been performed using the flow cytometer software to export counts of gated populations. The Acrosome 1 protocol using 3h incubation with compounds was run in 7 separate donor pools. The Acrosome 2 protocol using 3h incubation with compound followed by 1h stimulation of acrosome reaction was run in 5 separate donor pools.

**Sperm viability assay.** Cells were incubated with compounds for a total of 3h at 37°C. 20 min prior, propidium iodide was added (1.5 mM solution, Sigma, final dilution 1:2000). Plates

were analysed using a Satorius iQue Screener with settings to allow for a sampling time of ≤30 min per 384-well plate. Gating of data has been performed using the flow cytometer software to export counts of gated populations. This assay was tested in 5 separate donor pools.

**HeLa cell painting.** Cell were incubated with compounds for 24h. Cell painting was performed after [11] with modifications suggested in [68]:

- 10 µL of MitoTracker (Life Technologies, M22426, 1 mM stock in DMSO as recommended by the vendor) is added to 50 µL of media (1:2000 final dilution)

- Phalloidin-Alexa594 (Life Technologies, A12381, 66 µM stock in methanol as recommended by the vendor, 1:500 final dilution)

- WGA-Alexa594 (Life Technologies, W11262, 5 mg/mL stock, 1:250 final dilution) was used

- DAPI (5 mg/mL stock; 1:1000 final dilution) was used to stain DNA

- 20 µL of staining solution was used

- PBS was used for washing steps

**HepG2 cytotoxicity assay.** HepG2 assay was performed as described previously [64].

**Aqueous solubility.** The aqueous solubility of test compounds was measured using UHPLC, as previously described in [69].

**Data analysis.** For cell painting imaging data, plates of images were imported into OMERO [70]. Cell segmentation and feature extraction were performed using CellProfiler 3 [71] and using instructions from [11] to normalize data. Timelapse data of motility assays was processed as previously described [8]. Acrosome flow cytometry data was processed using NovoCyte Express (PerkinElmer), to gate out background debris, select for single cells and calculate percentage values of acrosome positive events. Sperm viability flow cytometry data was processed using iQue Forecyt (Satorius) to gate out background debris, select for single cells and calculate percentage values of propidium iodide positive events. For all sperm functional assays, data was normalized to DMSO control wells. Each plate had 16 DMSO wells. A median value of all 16 wells was used to calculated percentage change (pct_ch = ((value / DMSO_median) -1) * 100). For sperm viability we calculated fold-change (fc = 1 − (value / DMSO_median)). For each compound we calculated a p-value using a Welch's t-test implemented in scipy (scipy.stats.ttest_ind) comparing a compounds effect to DMSO controls.

The following packages have been used for data analysis and visualization:

R: Tidyverse, plotly, pHeatmap

Python: Pandas, NumPy, SciPy, Seaborn, Matplotlib, UMAP

## Supporting information

**S1 Fig. STB assay protocols.** (a) Diagram showing sperm motility assays run on high-content microscope with compound incubation time/read-out time. Motility assays were run under non-capacitating conditions, with a 1 hour recovery phase after preparation of spermatozoa. Capacitation assays were run under capacitating conditions. Spermatozoa are capacitated for 30 min prior to compound incubation. (b) Diagram showing assays run on high-throughput flow cytometer to measure acrosome status and sperm viability. Incubation time with compound, agonist and read-out times are indicated. Acrosome assays run under capacitating

conditions. Sperm viability assay run under non-capacitating condition. (c) Additional assays performed on sperm toolbox compounds.
(TIF)

**S2 Fig. Volcano plots of acrosome, capacitation and motility assays.** Upper panel indicates 10 μM, lower plot panels indicate 30 μM. Incubation times for assays are 3h with compound for Acrosome 2 assay and Motility (non-capacitating) (PM = progressive motility), and 30 min for the Capacitation assay (HA = hyperactive motility, PM = progressive motility). Dotted line indicates significance levels of 0.05. Any point above dotted line is indicated in magenta. Buffer conditions are indicated below Plot title (capacitating vs. non-capacitating).
(TIF)

**S1 Table. STB assay data.** Supporting data underlying the findings in this article.
(CSV)

## Acknowledgments

The authors thank Alistair Langlands, John Raynor and the UK National Phenotypic Screening Centre for support and assistance performing the automated screening assays.

## Author Contributions

**Conceptualization:** Kevin D. Read, Christopher L. R. Barratt, Ian H. Gilbert, Jason R. Swedlow.

**Data curation:** Anthony Richardson, Zoe C. Johnston, Rachel Myles, Neil R. Norcross.

**Formal analysis:** Franz S. Gruber.

**Funding acquisition:** Kevin D. Read, Sarah Martins da Silva, Christopher L. R. Barratt, Ian H. Gilbert, Jason R. Swedlow.

**Investigation:** Franz S. Gruber, Zoe C. Johnston, Neil R. Norcross, David P. Day, Irene Georgiou, Laura Sesma-Sanz, Caroline Wilson, Kevin D. Read.

**Methodology:** Franz S. Gruber, Zoe C. Johnston, Rachel Myles, Neil R. Norcross, David P. Day, Irene Georgiou, Laura Sesma-Sanz, Caroline Wilson.

**Project administration:** Christopher L. R. Barratt, Ian H. Gilbert, Jason R. Swedlow.

**Software:** Franz S. Gruber.

**Supervision:** Kevin D. Read, Sarah Martins da Silva, Christopher L. R. Barratt, Ian H. Gilbert, Jason R. Swedlow.

**Validation:** Franz S. Gruber, David P. Day, Irene Georgiou, Laura Sesma-Sanz, Caroline Wilson, Kevin D. Read.

**Visualization:** Franz S. Gruber, Jason R. Swedlow.

**Writing – original draft:** Franz S. Gruber, Zoe C. Johnston, Jason R. Swedlow.

**Writing – review & editing:** Franz S. Gruber, Anthony Richardson, Zoe C. Johnston, Kevin D. Read, Sarah Martins da Silva, Christopher L. R. Barratt, Ian H. Gilbert, Jason R. Swedlow.

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
