## [Decision Letter · Decision Letter 0]

15 Oct 2023

PONE-D-23-28149Sperm Toolbox – A selection of small molecules to study human spermatozoaPLOS ONE

Dear Dr. Swedlow,

Thank you for submitting your manuscript to PLOS ONE. After careful consideration, we feel that it has merit but does not fully meet PLOS ONE’s publication criteria as it currently stands. Therefore, we invite you to submit a revised version of the manuscript that addresses the points raised during the review process.

We look forward to receiving your revised manuscript.

Kind regards,

Wagdy Mohamed Eldehna, Ph.d

Academic Editor

PLOS ONE

Journal Requirements:

"The authors acknowledge support by the Bill & Melinda Gates Foundation (INV-007117)."

Reviewers' comments:

Reviewer's Responses to Questions

**Comments to the Author**

1. Is the manuscript technically sound, and do the data support the conclusions?

Reviewer #1: Partly

Reviewer #2: Yes

Reviewer #3: Yes

2. Has the statistical analysis been performed appropriately and rigorously? 

Reviewer #1: I Don't Know

Reviewer #2: I Don't Know

Reviewer #3: Yes

3. Have the authors made all data underlying the findings in their manuscript fully available?

Reviewer #1: Yes

Reviewer #2: Yes

Reviewer #3: Yes

4. Is the manuscript presented in an intelligible fashion and written in standard English?

Reviewer #1: Yes

Reviewer #2: Yes

Reviewer #3: Yes

5. Review Comments to the Author

Reviewer #1: 1- The paper exhibits strong writing and presents a well-conceived idea.

2- The inclusion of an online dataset is commendable.

3- I recommend incorporating all tested compounds (with numbering) into the manuscript for enhanced readability.

4- It would be beneficial to consolidate all results into tables based on their shared impact on sperm.

6- The absence of microscope images depicting actual sperm alterations is noteworthy; their inclusion would enhance the study.

7- The conclusion requires improvement as it lacks a concise summary of the main ideas.

Reviewer #2: The manuscript contains excellent work, and is well written except for some minor errors and few non scientific terms, that needs revision

Some issues should be addressed

1. Please clarify why 84 compounds only and what were the criteria of selection

2. The authors should consider performing enzyme assay for the identified network targets or at least some structure based insilico studies

3. Introduction is too short

4. The authors should discuss better the obtained results especially the importance of their findings and how it would help others

Reviewer #3: The authors present a relatively straightforward description of the construction of a new panel of compounds that can further the study of both infertility and contraception. The study builds upon earlier work from the same group in which they have set up a high-throughput phenotypic screening platform. The work is valuable for people in the field as it presents a new compound panel that can be used to support screening.

I have no reservations regarding the scientific value of the experiments and data collected. Yet the paper could use some clarification in relation to two aspects: i) It is not particularly clear what the precise or relative value of the selected compound set is. And I would like to read some reflection on that because in the end the set is the deliverable that deserves reporting; and ii) the text lacked some introduction to the methods which made it harder to follow the reasoning and distinguishing observation from interpretation.

detailed comments:

i)

Various selection criteria were used to obtain the final set of screening compounds. To evaluate the relative power/value of this specific set it is important to evaluate the contribution of the individual compounds (i.e. could the set have been made smaller without losing much resolution) and at the same time the potential contribution of compounds that were left out (i.e. the resolution that still could have been gained). It is necessary to read a bit more argued reflection on the added value of the presented set in the paper.

In that light, I have a conceptual problem with l 123 - 126 because the paper states that some included compounds hardly showed effects. Yet they were not removed because they were reported earlier in literature. But why then test this list of compounds at all, they were all already described in literature, as that was a selection criterion? And, at the other end, are all active compounds already known? Why have other compounds not been tested? Phrased in other words: Can the authors indicate the relative value of the panel at all, when following their line of reasoning?

ii)

l 80 and following. It would help if the various tests were shortly introduced, maybe shortly describe the platform. Because such a short introduction to the available tests (within the existing platform, I assume) lacks, it is not really clear what the paragraphs are precisely about. e.g. l 95 starts with: "the acrosome assays" Given the text uptill then, the reader is not informed what the assay is about and why it is used.

text

l 73 are -> were

l 74 indicate -> indicated; are -> were

l 77 consists -> consisted

l 83 consists -> consisted

l 85 and normalized -> What is meant here? Was the outcome compared to a control (substraction) or were they really normalized (division/multiplication).

in l 108 it is not clear whether it states an observation (then it should have been in past tense) or a conclusion (but then I would like to read some arguments).

l 123 show -> showed

6. PLOS authors have the option to publish the peer review history of their article (what does this mean?). If published, this will include your full peer review and any attached files.

Reviewer #1: **Yes: **Abdulrahman M. Saleh

Reviewer #2: **Yes: **Mahmoud A. El Hassab

Reviewer #3: No

---

## [Author Response · Author response to Decision Letter 0]

15 Dec 2023

Dear Dr Eldehna

Thank you very much for the reviews our paper describing the Sperm Toolbox as a resource for the research and drug discovery community. We appreciate the comments and insights from the reviewer. 

We have separated each of the comments from the reviewers and provided responses in a table in the Response to Reviewers document. We hope the comments and the accompanying ms edits satisfy the requests from the reviewers.

We look forward to your comments and responses. 

For the authors, 

Jason Swedlow

---

## [Decision Letter · Decision Letter 1]

11 Jan 2024

Sperm Toolbox – A selection of small molecules to study human spermatozoa

PONE-D-23-28149R1

Dear Dr. Swedlow,

We’re pleased to inform you that your manuscript has been judged scientifically suitable for publication and will be formally accepted for publication once it meets all outstanding technical requirements.

Kind regards,

Wagdy Mohamed Eldehna, Ph.d

Academic Editor

PLOS ONE

Additional Editor Comments (optional):

Reviewers' comments:

Reviewer's Responses to Questions

**Comments to the Author**

1. If the authors have adequately addressed your comments raised in a previous round of review and you feel that this manuscript is now acceptable for publication, you may indicate that here to bypass the “Comments to the Author” section, enter your conflict of interest statement in the “Confidential to Editor” section, and submit your "Accept" recommendation.

Reviewer #1: All comments have been addressed

2. Is the manuscript technically sound, and do the data support the conclusions?

Reviewer #1: Yes

3. Has the statistical analysis been performed appropriately and rigorously? 

Reviewer #1: I Don't Know

4. Have the authors made all data underlying the findings in their manuscript fully available?

Reviewer #1: Yes

5. Is the manuscript presented in an intelligible fashion and written in standard English?

Reviewer #1: Yes

6. Review Comments to the Author

Reviewer #1: All required modifications were completed well

Therefore, I recommend accepting this article in its current form.

7. PLOS authors have the option to publish the peer review history of their article (what does this mean?). If published, this will include your full peer review and any attached files.

Reviewer #1: **Yes: **Abdulrahman M. Saleh

---

## [Editor Report · Acceptance letter]

10 Feb 2024

PONE-D-23-28149R1 

PLOS ONE

Dear Dr. Swedlow, 

I'm pleased to inform you that your manuscript has been deemed suitable for publication in PLOS ONE. Congratulations! Your manuscript is now being handed over to our production team.

Kind regards, 

on behalf of

Dr. Wagdy Mohamed Eldehna 

Academic Editor

PLOS ONE